# Deep Dynamical Modeling and Control of Unsteady Fluid Flows

**Jeremy Morton**[*]
jmorton2@stanford.edu

**Freddie D. Witherden**[*]
fdw@stanford.edu

**Antony Jameson** [†]
antony.jameson@tamu.edu

**Mykel J. Kochenderfer**[*]
mykel@stanford.edu

## Abstract

The design of flow control systems remains a challenge due to the nonlinear nature of the equations that govern fluid flow. However, recent advances in computational fluid dynamics (CFD) have enabled the simulation of complex fluid flows with high accuracy, opening the possibility of using learning-based approaches to facilitate controller design. We present a method for learning the forced and unforced dynamics of airflow over a cylinder directly from CFD data. The proposed approach, grounded in Koopman theory, is shown to produce stable dynamical models that can predict the time evolution of the cylinder system over extended time horizons. Finally, by performing model predictive control with the learned dynamical models, we are able to find a straightforward, interpretable control law for suppressing vortex shedding in the wake of the cylinder.

## 1 Introduction

Fluid flow control represents a significant challenge, with the potential for high impact in a variety of sectors, most notably the automotive and aerospace industries. While the time evolution of fluid flows can be described by the Navier-Stokes equations, their nonlinear nature means that many control techniques, largely derived for linear systems, prove ineffective when applied to fluid flows. An illuminating test case is the canonical problem of suppressing vortex shedding in the wake of airflow over a cylinder. This problem has been studied extensively experimentally and computationally, with the earliest experiments dating back to the 1960s [1]–[6]. These studies have shown that controller design can prove surprisingly difficult, as controller effectiveness is highly sensitive to flow conditions, measurement configuration, and feedback gains [3], [5]. Nonetheless, at certain flow conditions vortex suppression can be achieved with a simple proportional control law based on a single sensor measurement in the cylinder wake. Thus, while the *design* of flow controllers may present considerable challenges, effective controllers may in fact prove relatively easy to *implement*.

Recent advances in computational fluid dynamics (CFD) have enabled the numerical simulation of previously intractable flow problems for complex geometries [7]–[11]. Such simulations are generally run at great computational expense, and generate vast quantities of data. In response, the field of reduced-order modeling (ROM) has attracted great interest, with the aim of learning efficient dynamical models from the generated data. This research has yielded a wide array of techniques for learning data-driven dynamical models, including balanced truncation, proper orthogonal decomposition, and dynamic mode decomposition [12]. Recent work has sought to incorporate reduced-order models with robust- and optimal-control techniques to devise controllers for nonlinear systems [6], [13], [14].

In parallel, the machine learning community has devoted significant attention to learning-based control of complex systems. Model-free control approaches attempt to learn control policies without

---

[*]Department of Aeronautics and Astronautics, Stanford University
[†]Department of Aerospace Engineering, Texas A&M University

constructing explicit models for the environment dynamics, and have achieved impressive success in a variety of domains [15], [16]. However, model-free methods require many interactions with an environment to learn effective policies, which may render them infeasible for many flow control applications where simulations are computationally expensive. In contrast, model-based control approaches have the potential to learn effective controllers with far less data by first modeling the environment dynamics. Model-based methods have been successful in some domains [17], [18], but such success hinges on the need to construct accurate dynamical models.

In this work, we apply recent advances in the fields of reduced-order modeling and machine learning to the task of designing flow controllers. In particular, we extend an algorithm recently proposed by Takeishi *et al.* [19], leveraging Koopman theory to learn models for the forced and unforced dynamics of fluid flow. We show that this approach is capable of stably modeling two-dimensional airflow over a cylinder for significant prediction horizons. We furthermore show that the learned dynamical models can be incorporated into a model predictive control framework to suppress vortex shedding. Finally, we discuss how the actions selected by this controller shed insight on a simple and easy-to-implement control law that is similarly effective.While applied to fluid flow in this work, we note that the proposed approach is general enough to be applied to other applications that require modeling and control of high-dimensional dynamical systems.

## 2    Modeling unforced dynamics

Let $x_t \in \mathbb{R}^n$ be a state vector containing all data from a single time snapshot of an unforced fluid flow simulation. Our goal is to discover a model of the form $x_{t+1} = F(x_t)$ that describes how the state will evolve in time. The function $F$ can take many forms; one possibility is that the system dynamics are linear, in which case the state updates will obey $x_{t+1} = Kx_t$, with $K \in \mathbb{R}^{n \times n}$. If we observe a sequence of time snapshots $x_{1:T+1}$, we can construct the matrices

$$X = [x_1, x_2, \ldots, x_T] \quad \text{and} \quad Y = [x_2, x_3, \ldots, x_{T+1}] \tag{1}$$

and subsequently find the matrix $A = YX^{\dagger}$, where $X^{\dagger}$ is the Moore-Penrose pseudoinverse of $X$. As $T$ increases, $A$ will asymptotically approach $K$ [19], and hence approximate the true system dynamics. Such an approximation will in general only be accurate for systems with linear dynamics; in the following section we discuss how a similar approximation can be formed for nonlinear systems.

### 2.1    The Koopman operator

Consider a nonlinear discrete-time dynamical system described by $x_{t+1} = F(x_t)$. Furthermore, let the Koopman operator $\mathcal{K}$ be an infinite-dimensional linear operator that acts on all observable functions $g : \mathbb{R}^n \to \mathbb{C}$. Koopman theory asserts that a nonlinear discrete-time system can be mapped to a *linear* discrete-time system, where the Koopman operator advances observations of the state forward in time [20]:

$$\mathcal{K}g(x_t) = g(F(x_t)) = g(x_{t+1}). \tag{2}$$

While Koopman theory provides a lens under which nonlinear systems can be viewed as linear, its applicability is limited by the fact that the Koopman operator is infinite-dimensional. However, if there exist a finite number of observable functions $\{g_1, \ldots, g_m\}$ that span a subspace $\mathcal{G}$ such that $\mathcal{K}g \in \mathcal{G}$ for any $g \in G$, then $\mathcal{G}$ is considered to be an invariant subspace and the Koopman operator becomes a finite-dimensional operator $K$. We abuse notation by defining the vector-valued observable $g = [g_1, \ldots, g_m]^{\mathsf{T}}$, and furthermore define the matrices

$$\tilde{X} = [g(x_1), g(x_2), \ldots, g(x_T)] \quad \text{and} \quad \tilde{Y} = [g(x_2), g(x_3), \ldots, g(x_{T+1})]. \tag{3}$$

The matrix $A = \tilde{Y}\tilde{X}^{\dagger}$ will asymptotically approach the Koopman operator $K$ with increasing $T$. Takeishi *et al.* showed that the task of finding a set of observables that span an invariant subspace reduces to finding a state mapping $g(x_t)$ under which linear least-squares regression performs well (i.e. the loss $\|\tilde{Y} - (\tilde{Y}\tilde{X}^{\dagger})\tilde{X}\|_F^2$ is minimized), and proposed learning such a mapping with deep neural networks [19]. In experiments, their proposed algorithm was shown to perform well in analysis and prediction on a number of low-dimensional dynamical systems.

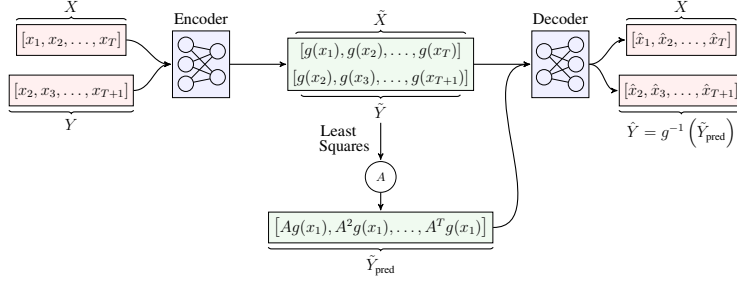

Figure 1: Illustration of procedure used to train Deep Koopman dynamical models.

## 2.2 Deep Koopman dynamical model

We now present the Deep Koopman model, which employs a modified form of the training algorithm proposed by Takeishi *et al.* to learn state mappings that approximately span a Koopman invariant subspace. The training algorithm is depicted in Fig. 1. First, a sequence of time snapshots $x_{1:T+1}$ is used to construct the matrices $X$ and $Y$ defined in Eq. (1). These matrices are fed into an encoder neural network, which serves as the mapping $g(x_t)$ and produces the matrices $\tilde{X}$ and $\tilde{Y}$ defined in Eq. (3). Subsequently, a linear least-squares fit is performed to find an $A$-matrix that can propagate the state mappings forward in time. Finally, $\tilde{X}$ and the propagated state mappings are fed into a decoder that functions as $g^{-1}$ to yield the matrices $\hat{X}$ and $\hat{Y}$, approximations to $X$ and $Y$.

The Deep Koopman model is trained to minimize $\mathcal{L} = \|X - \hat{X}\|_F^2 + \|Y - \hat{Y}\|_F^2$, where $\hat{Y}$ is obtained by running $\tilde{Y}_{\text{pred}}$ through the decoder. Minimizing the error between $X$ and $\hat{X}$ enforces that the mapping $g(x_t)$ is invertible, while minimizing the error between $Y$ and $\hat{Y}$ enforces that the derived dynamical model can accurately simulate the time evolution of the system. One main difference between our algorithm and that proposed by Takeishi *et al.* is that we force the model to simulate the time evolution of the system during training in a manner that mirrors how the model will be deployed at test time. In particular, we apply the derived $A$-matrix recursively to state mapping $g(x_1)$ to produce the matrix $\tilde{Y}_{\text{pred}}$ defined in Fig. 1, which is then mapped to $\hat{Y}$ through the decoder.

To better clarify another new feature of our proposed algorithm, it is worth drawing a distinction between *reconstruction* and *prediction*. If a dynamical model is constructed based on a sequence of states $x_{1:N}$, then simulations generated by the dynamical model would be *reconstructing* the already observed time evolution of the system for all time steps $t \leq N$, and *predicting* the time evolution of the system for all time steps $t > N$. We would like to train a dynamical model that we can ultimately use to predict the future time evolution of a given system. Thus, during training we generate the $A$-matrix based on only the first $T/2$ entries of $\tilde{X}$ and $\tilde{Y}$, thereby enforcing that the last $T/2$ entries of $\tilde{Y}_{\text{pred}}$ are purely predictions for how the system will evolve in time.

One of the advantages of this approach is its relative simplicity, as the neural network architecture is equivalent to that of a standard autoencoder. The dynamics matrix $A$ does not need to be modeled directly; rather, it is derived by performing least squares on the learned state mappings. In our implementation, the encoder consists of ResNet convolutional layers [21] with ReLU activations followed by fully connected layers, while the decoder inverts all operations performed by the encoder. We applied $L_2$ regularization to the weights in the encoder and decoder. The gradients for all operations are defined in Tensorflow [22], and the entire model can be trained end-to-end to learn suitable state mappings $g(x_t)$.

## 2.3 Experiments

We now provide a description for how we train and evaluate the Deep Koopman models. We first describe the test case under study, then quantify the ability of the Deep Koopman model to learn the system dynamics.

### 2.3.1 Test case

The system under consideration is a two dimensional circular cylinder at Reynolds number 50 and an effectively incompressible Mach number of 0.2. This is a well-studied test case [23], [24] which

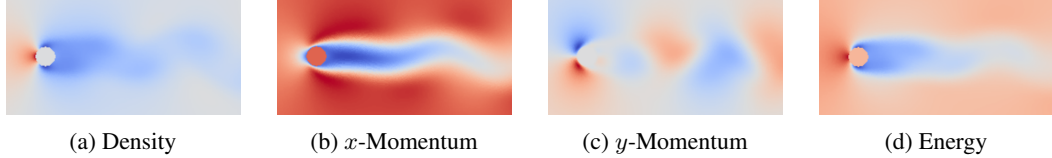

(a) Density          (b) $x$-Momentum          (c) $y$-Momentum          (d) Energy

Figure 2: Format of inputs to neural network. Different physical quantities are treated as different channels in the input.

has been used extensively both for validation purposes and as a legitimate research case in its own right. The chosen Reynolds number is just above the cut-off for laminar flow and thus results in the formation of a von Karman vortex street, where vortices are shed from the upper and lower surface of the cylinder in a periodic fashion. Vortex shedding gives rise to strong transverse forces, and is associated with higher drag and unsteady lift forces [2], [6].

To perform the fluid flow simulations, we use a variation of the high-order accurate PyFR solver [25].The surface of the cylinder is modeled as a no-slip isothermal wall boundary condition, and Riemann invariant boundary conditions are applied at the far-field. The domain is meshed using 5672 unstructured, quadratically curved quadrilateral elements. All simulations are run using quadratic solution polynomials and an explicit fourth order Runge–Kutta time stepping scheme.

A training dataset is constructed by saving time snapshots of the system every 1500 solver steps. To make the simulation data suitable for incorporation into a training algorithm, the data is formatted into image-like inputs before storage. Solution quantities are sampled from a $128 \times 256$ grid of roughly equispaced points in the neighborhood of the cylinder. Each grid point contains four channels corresponding to four physical quantities in the flow at that point: density, $x$-momentum, $y$-momentum, and energy. An example snapshot can be found in Fig. 2, illustrating the qualitative differences between the four distinct input channels.

### 2.3.2   Deep Variational Bayes Filter

We compare the Deep Koopman model with the Deep Variational Bayes Filter (VBF) [26] to baseline performance. The Variational Bayes Filter can be viewed as a form of *state space model*, which seeks to map high-dimensional inputs $x_t$ to lower-dimensional latent states $z_t$ that can be evolved forward in time. The VBF is a recently proposed approach that improves upon previous state space models (e.g. Embed to Control [17]) and evolves latent states forward linearly in time, and thus serves as a suitable performance benchmark for the Deep Koopman model.

We use the same autoencoder architecture employed by the Deep Koopman model to perform the forward and inverse mappings between the inputs $x_t$ and the latent states $z_t$. As with the Deep Koopman model, the inputs are time snapshots from CFD simulations. The time evolution of the latent states is described by $z_{t+1} = A_t z_t + B_t u_t + C_t w_t$, where $u_t$ is the control input at time $t$ and $w_t$ represents process noise. The matrices $A_t$, $B_t$, and $C_t$ are assumed to comprise a locally linear dynamical model, and are determined at each time step as a function of the current latent state and control input. Since we seek to model the unforced dynamics of the fluid flow, we ignore the effect of control inputs in our implementation of the Deep VBF.

### 2.3.3   Results

In addition to the Deep VBF, we also benchmark the Deep Koopman model against a model trained using the procedure proposed by Takeishi *et al.*, which sets $\tilde{Y}_{\text{pred}} = A\tilde{X}$ rather than calculating $\tilde{Y}_{\text{pred}}$ by applying $A$ recursively to $g(x_1)$. Each model is trained on 32-step sequences of data extracted from two-dimensional cylinder simulations. We then use the trained models to recreate the time evolution of the system observed during 20 test sequences and extract the error over time. For a fair comparison, $g(x_t)$ and $z_t$ are defined to be 32-dimensional vectors. The Koopman method construct its dynamical model based on state mappings from the first 16 time steps, then simulates the system for all time steps using the derived $A$-matrix. The Takeishi baseline derives its $A$-matrix based on state mappings from the first 32 time steps. The VBF constructs a new locally linear dynamical model at each time step, but relies on information from the first 32 time steps to sample the initial value of the process noise $w_0$.

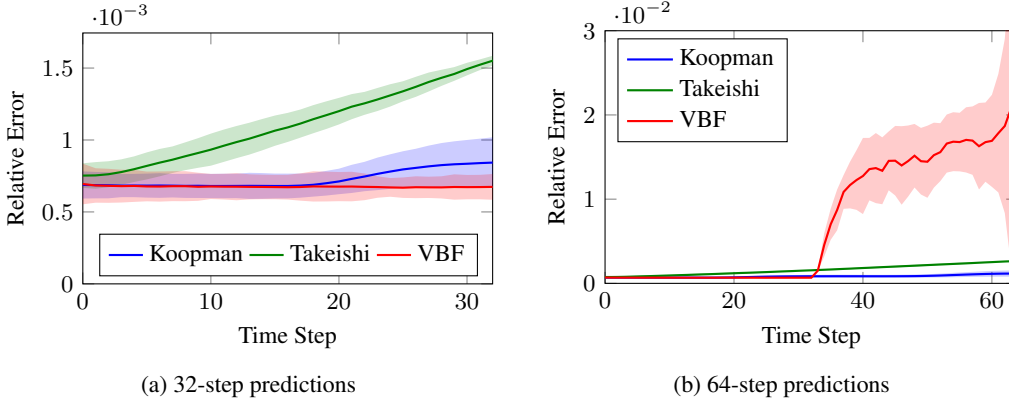

(a) 32-step predictions

(b) 64-step predictions

Figure 3: Average prediction errors over time for Deep Koopman and Deep Variational Bayes Filter models. Solid lines represent the mean prediction error across 20 sequences, while the shaded regions correspond to one standard deviation about the mean.

The results of these experiments can be found in Fig. 3a and Fig. 3b, where the error metric is the relative error, defined as the $L_1$-norm of the prediction error normalized by the $L_1$-norm of the ground-truth solution. We can see in Fig. 3a that the error for the Takeshi baseline initially grows more rapidly than the error for the other models. This illustrates the importance of training models to generate recursive predictions, since models trained to make single-step predictions tend to generate poor multi-step predictions due to prediction errors compounding over time [27]. Over a 32-step horizon the Deep Koopman and VBF models perform comparably, with the error for the Koopman model rising slightly at later time steps as it begins generating predictions for states that it did not have access to in constructing its dynamical model. A much starker contrast in model performance can be observed in Fig. 3b, where the Variational Bayes Filter begins to rapidly accumulate error once it surpasses the 32-step time horizon.

For the results shown in Fig. 3b, the Variational Bayes Filter is performing reconstruction for 32 steps and prediction for 32 steps. Hence, we see that the VBF is effective at reconstruction, but is unable to function stably in prediction. In contrast, we see that the Deep Koopman model, aided by its ability to construct state mappings that approximately span an invariant subspace, is able to generate stable predictions for much longer time horizons. In fact, while the prediction error of the Koopman model does grow with time, its mean prediction error remains less than $0.2\%$ over a horizon of 128 time steps, corresponding to approximately eight periods of vortex shedding. Since we ultimately want a predictive dynamical model that can be incorporated into a control framework, we conclude that the Deep Koopman model is well suited for the task.

## 3 Modeling forced dynamics

We now explain how the proposed Deep Koopman algorithm can be extended to account for the effect that control inputs have on the time evolution of dynamical systems.

### 3.1 Deep Koopman model with control

We have already demonstrated how the Deep Koopman algorithm can learn state mappings that are suitable for modeling unforced dynamics. In accounting for control inputs, we now aim to construct a linear dynamical model of the form $g(x_{t+1}) = Ag(x_t) + Bu_t$, where $u_t \in \mathbb{R}^m$ and $B \in \mathbb{R}^{n \times m}$. Defining the matrix $\Gamma = [u_1, \dots, u_T]$, we would like to find a dynamical model $(A, B)$ that satisfies $\tilde{Y} = A\tilde{X} + B\Gamma$. Proctor *et al.* presented several methods for estimating $A$ and $B$ given matrices $\tilde{X}$ and $\tilde{Y}$ [28]. In this work, we choose to treat $B$ as a known quantity, which means that $A$ can be estimated through a linear least-squares fit

$$A = (\tilde{Y} - B\Gamma)\tilde{X}^{\dagger}. \tag{4}$$

Thus, the Deep Koopman training algorithm presented in Section 2.2 can be modified such that $A$ is generated through Eq. (4). While we treat $B$ as a known quantity, in reality it is another parameter that we must estimate. We account for this by defining a global $B$-matrix, whose paramaters are optimized by gradient descent during training along with the neural network parameters.

## 3.2 Modified test case

With the ability to train Koopman models that account for control inputs, we now consider a modified version the two-dimensional cylinder test case that allows for a scalar control input to affect the fluid flow. In particular, the simulation is modified so that the cylinder can rotate with a prescribed angular velocity. Cylinder rotation is modeled by applying a spatially varying velocity to the surface of the wall, thus enabling the grid to remain static. The angular velocity is allowed to vary every 1500 solver steps, with the value held constant during all intervening steps.

## 3.3 Training process

We train Koopman models on data from the modified test case to construct models of the forced dynamics. A training dataset is collected by simulating the two-dimensional cylinder system with time-varying angular velocity. Every 1500 solver steps, a time snapshot $x_t$ is stored and the control input $u_t$ is altered. In total, the training set contains 4238 snapshots of the system. We then divide these snapshots into 1600 staggered 32-step sequences for training the Deep Koopman model. As in the case of unforced dynamics, during training dynamical models are constructed based on information from the first 16 time steps, but the system is simulated for 32 time steps. Training a single model takes approximately 12 hours on a Titan X GPU.

The form of control inputs applied to the system in generating the training data has a strong effect on the quality of learned models. Analogous to frequency sweeps in system identification [29], we subject the system to sinusoidal inputs with a frequency that increases linearly with time. These sinusoidal inputs are interspersed with periods with no control inputs to allow the model to learn the unforced system dynamics from different initial conditions.

# 4 Model predictive control

We evaluate the quality of the learned Deep Koopman models by studying their ability to enable effective control of the modeled system. In particular, we incorporate the learned dynamical models into a model predictive control (MPC) framework with the aim of suppressing vortex shedding. At each time step in MPC, we seek to find a sequence of inputs that minimizes the finite-horizon cost:

$$J_T = \sum_{t=1}^{T} (c_t - c_{\text{goal}})^\intercal Q (c_t - c_{\text{goal}}) + \sum_{t=1}^{T-1} R u_t^2, \tag{5}$$

where $c_t = g(x_t)$ represents the observable of state $x_t$, $c_{\text{goal}} = g(x_{\text{goal}})$ represents the observable of goal state $x_{\text{goal}}$, $Q$ is a positive definite matrix penalizing deviation from the goal state, and $R$ is a nonnegative scalar penalizing nonzero control inputs. We can furthermore apply the constraints $|u_t| < u_{\max} \ \forall \ t$, $c_1 = g(x_1)$, and $c_{t+1} = A c_t + B u_t$ for $t = 2, \ldots, T$, where $A$ and $B$ are generated by the Deep Koopman model. As formulated this optimization problem is a quadratic program, which can be solved efficiently with the CVXPY software [30].

For $x_{\text{goal}}$ we use a snapshot of the steady flow observed when the cylinder system is simulated at a Reynolds number of 45, which is a sufficiently low Reynolds number that vortex shedding does not occur. While the flow at this lower Reynolds number is qualitatively different from the flow at a Reynolds number of 50, we find that formulating the problem in this way leads to a reliable estimate of the cost, as demonstrated in the next section.

We use an MPC horizon of $T = 16$ time steps. This aligns with the Deep Koopman model training process, where the neural network generates predictions for 16 time steps *beyond* what it uses to construct its dynamical model. At each time step, we find state mappings for the previous 16 time steps, and use those mappings in conjunction with the global $B$-matrix to find a suitable $A$-matrix for propagating the state mappings forward in time. We then solve the optimization problem described by Eq. (5) to find $u_{1:T}^*$, the optimal sequence of control inputs. We set $Q = I$, the identity matrix, and $R \sim 10^5$, which accounts for the fact that $\|c_t - c_{\text{goal}}\|_2$ is typically orders of magnitude larger than $|u_t|$ and discourages actions that are too extreme for the dynamical model to handle accurately. The first input, $u_1^*$, is passed to the CFD solver, which advances the simulation forward in time.

## 4.1 MPC results

We now present the results of performing model predictive control on the two-dimensional cylinder problem. To evaluate the effectiveness of the derived controller, we require a measure of closeness

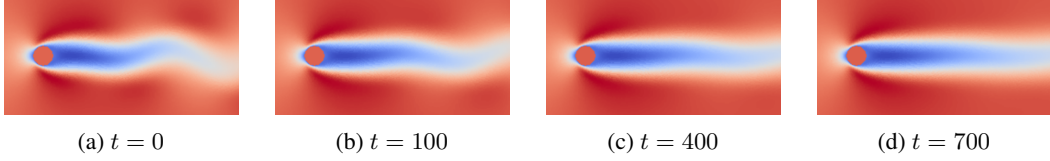

(a) $t = 0$       (b) $t = 100$       (c) $t = 400$       (d) $t = 700$

Figure 4: Snapshots of $x$-momentum over time as the MPC algorithm attempts to suppress vortex shedding.

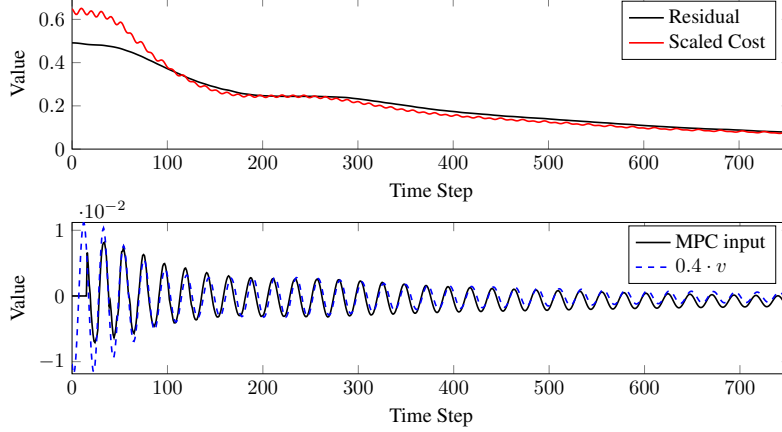

Figure 5: Above: scaled residuals plotted alongside estimated cost used for MPC action selection. Below: control inputs selected by MPC plotted along with scaled $y$-velocity measurements.

to the desired outcome; in this case, this desired outcome is to achieve a steady laminar flow devoid of vortex shedding. The Navier–Stokes equations are a conservation law, taking the form of $\frac{\partial q}{\partial t} = -\boldsymbol{\nabla} \cdot \mathbf{f}(q, \boldsymbol{\nabla}q)$, where $q = [\rho, \rho u, \rho v, E]$ is a vector of the density, $x$-momentum, $y$-momentum, and energy fields respectively, and $\mathbf{f}(q, \boldsymbol{\nabla}q)$ is a suitably defined flux function. In the process of running CFD simulations, PyFR evaluates the right-hand side of this equation by calculating residuals. Note that if a steady flow is achieved, the time derivative will be zero and in turn the residuals will be zero. Thus, we use residual values as a measure of closeness to the desired steady flow. In particular, we extract the norm of the residuals for $x$- and $y$-momentum over time.

Results from the model predictive control experiments can be found in Fig. 4 and Fig. 5. In Fig. 4, we get a qualitative picture for the effectiveness of the applied control, as the cylinder wake exhibits a curved region of low $x$-momentum characteristic of vortex shedding at early time steps, then flattens out to a profile more characteristic of laminar flow over time. In the upper plot of Fig. 5, we get a quantitative picture of the controller performance, showing that the controller brings about a monotonic decrease in the residuals over time. Additionally, we plot a scaled version of $\|g(x_t) - g(x_{\text{goal}})\|_2$. Interestingly, we note that there is a strong correspondence between a decrease in the residuals and a decrease in the cost that model predictive control is attempting to minimize. This provides confidence that using this measure of cost in MPC is sensible for this problem.

The lower plot in Fig. 5 shows the control inputs applied to the system over time. A dynamical model cannot be constructed until 16 states have been observed, so the control inputs are initially set to zero. Subsequently, the inputs appear to vary sinusoidally and decrease in amplitude over time. Remarkably, it is possible to find a location in the wake of the cylinder, at a location denoted by $d^*$, where the variations in $y$-velocity are in phase with the selected control inputs. When scaled by a constant value of 0.4, we see a strong overlap between the control inputs and velocity values. Viewed in this light, we note that the controller obtained through MPC is quite interpretable, and is functionally similar to a proportional controller performing feedback based on $y$-velocity measurements with a gain of 0.4.

### 4.2 Proportional control

With the insights gained in the previous section, we now test the effectiveness of a simple proportional control scheme in suppressing vortex shedding. Rather than selecting inputs with MPC, we set the angular velocity of the cylinder by applying a gain of 0.4 to measurements of $y$-velocity at point $d^*$.

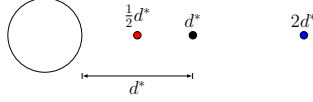

Figure 6: Schematic of measurement points for performing proportional control.

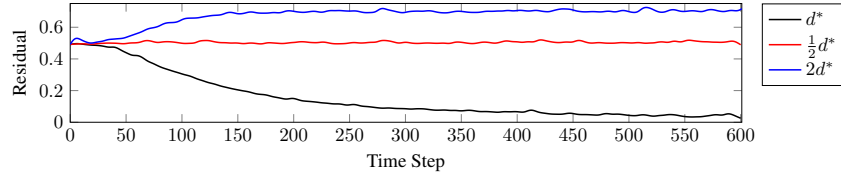

Figure 7: Calculated residuals over time for different measurement locations.

While easy to implement, we perform additional experiments to illustrate that such a control law is not easy to find. In these experiments, we attempt to perform proportional control with the same gain based on measurements at two additional locations, $\frac{1}{2}d^*$ and $2d^*$, as illustrated in Fig. 6.

The results of these experiments, summarized in Fig. 7, demonstrate that proportional control based on measurements at $d^*$ is effective at suppressing vortex shedding. Meanwhile, proportional control laws based on measurements at the other locations are unable to drive the system closer to the desired steady, laminar flow. These results are in agreement with previous studies [3], which showed that the effectiveness of proportional control is highly dependent upon the measurement location.

## 5 Related work

While originally introduced in the 1930s [31], the Koopman operator has attracted renewed interest over the last decade within the reduced-order modeling community due to its connection to the dynamic mode decomposition (DMD) algorithm [20], [32]. DMD finds approximations to the Koopman operator under the assumption that the state variables $x$ span an invariant subspace. However, they will not span an invariant subspace if the underlying dynamics are nonlinear. Extended DMD (eDMD) approaches build upon DMD by employing a richer set of observables $g(x)$, which typically need to be specified manually [33]. A number of recent works have studied whether deep learning can be used to learn this set of observables automatically, thereby circumventing the need to hand-specify a dictionary of functions [19], [34]–[36].

In the machine learning community, recent work focused on learning deep dynamical models from data has showed that these models can enable more sample-efficient learning of effective controllers [37]–[39]. Our work most closely parallels work in state representation learning (SRL) [40], which focuses on learning low-dimensional features that are useful for modeling the time evolution of high-dimensional systems. Recent studies have worked toward learning state space models from image inputs for a variety of tasks, and have been designed to accommodate stochasticity in the dynamics and measurement noise [17], [26], [41]. Given that both Koopman-centric approaches and SRL attempt to discover state mappings that are useful for describing the time evolution of high-dimensional systems, an opportunity likely exists to bridge the gap between these fields.

## 6 Conclusions

We introduced a method for training Deep Koopman models, demonstrating that the learned models were capable of stably simulating airflow over a cylinder for significant prediction horizons. Furthermore, we detailed how the Koopman models could be modified to account for control inputs and thereby leveraged for flow control in order to suppress vortex shedding. Learning sufficiently accurate dynamical models from approximately 4000 training examples, the method is very sample efficient, which is of high importance due to the large computational cost associated with CFD simulations. Most importantly, by incorporating the Deep Koopman model into an MPC framework, we showed that the resulting control law was both interpretable and sensible, aligning with well studied flow control approaches from the literature. Future work will focus on applying the proposed approach to flows at higher Reynolds numbers to see how its effectiveness scales to increasingly complex flows. Furthermore, we hope to apply the proposed approach to other flow control problems, studying whether it can provide similar insight into how to design controllers for other applications. The code associated with this work can be found at `https://github.com/sisl/deep_flow_control`.

**Acknowledgments**

The authors would like to thank the reviewers for their insightful feedback. This material is based upon work supported by the National Science Foundation Graduate Research Fellowship Program under Grant No. DGE- 114747. The authors would like to thank the Air Force Office of Scientific Research for their support via grant FA9550-14-1-0186.

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
