[Reviews · NeurIPS 2018]

Reviewer 1



Update: after author response. I think that the authors have done good job in addressing my concerns in their response. Prior to receiving the feedback, my main concern could be summarized as: the paper proposes a modification to an existing method (optimize multi-step prediction error instead of one-step prediction error) in an effort to improve accuracy and stability of the model during open-loop simulations. However, there was no direct comparison to the existing method that was modified, and so in the absence of any theoretical or empirical evidence, it is difficult to say whether the proposed approach is effective. In their response, the authors provided a comparison with Takeishi, which suggests that the modification does indeed (at least in this particular experiment) have the desired effect of improving multi-step prediction accuracy. In light of these results (and with the cost function specified more clearly) I'm happy to revise my 'score' to a 6. Summary ------- This paper concerns data-driven modeling and control of fluid flows. The dynamics describing such flows are nonlinear, making modeling and control challenging. The approach adopted in this paper is based on spectral analysis of the Koopman operator (a popular approach that has received considerable attention in the literature), which (roughly speaking) models the finite-dimensional nonlinear dynamics with infinite-dimensional linear dynamics. For tractable analysis it is desirable to find a finite-dimensional approximation of these infinite-dimensional dynamics (i.e., a Koopman invariant subspace). The strategy presented in this paper is dynamic mode decomposition (DMD), another popular approach that has received much attention in the literature. The key challenge in DMD is coming up with a basis for the Koopman invariant subspace (i.e., a collection of nonlinear functions referred to as ‘observables’). Here lies the main contribution of the paper: a data-driven way to learn appropriate observables, where each observable function is modeled by a deep neural network (encoder). The resulting finite-dimensional linear model is then used for control design (standard MPC for linear systems). It is observed that the MPC strategy coincides with proportional feedback (i.e. control action proportional to the velocity), *as long as* the velocity used for feedback is measured at a specific point (spatially). This proportional control then constitutes a ‘simple, interpretable’ policy. Originality ----------- The idea of learning a basis for the Koopman invariant subspace (using deep neural networks) seems very promising. Such an approach was proposed in Takeishi (NIPS ’17, ref [19]). As the authors themselves state, the approach in this paper is a ‘modified form of the training algorithm proposed by Takeishi’, c.f., pg. 2 - this raises the question: how significant are the modifications? In my view, the modifications are rather minor. in the authors’ own words: ‘One main difference between our algorithm and that proposed by Takeishi et al. is that we force the model to simulate the time evolution of the system during training in a manner that mirrors how the model will be deployed at test time.’ Specifically, the error term for the decoder includes mismatch for both the measured states and the predicted states (where the predicted states are obtained by simulating the linear model forward in time, recursively). While this does not directly effect the estimation of A (which is estimated by least squares) the argument is (presumably) that this indirectly (via the nonlinear function g) leads to better prediction performance (specifically, longer-term stability of predictions, c.f. e.g., Figure 3). A few comments: i) while it is plausible that this could improve predictive performance, it is not obvious; e.g., this does not directly influence the estimation of A (which is still fit by least squares, i.e., minimization of one-step prediction error), and it is A that affects stability of the long-term predictions. ii) in my view, this is a rather minor modification of the algorithm proposed in Takeishi. iii) further, this appears to be the only difference? (i.e., ‘one main difference’ could more accurately be described as ‘the only difference’?). The other difference is that Takeishi has an additional step to estimate a linear delay embedding (because Takeishi do not assume that the states are observable, unlike the present paper, which makes the stronger assumption that the states are observable). The material on MPC is entirely standard, given the finite-dimensional linear model. Quality ------- There are no theoretical claims made in the paper, so quality can only be assessed via the numerical illustrations. The method is evaluated against a 'deep variational Bayes filter'. A natural question is: why not compare performance to the method proposed by Takeishi? After all, the proposed method is a modification of Takeishi, would it not make sense to see if the modification had the intended effect, namely, better long-term predictive performance and stability? Clarity ------- The paper is fairly well written, however, the main contribution (namely, a method for training deep Koopman dynamical models) is only really presented via Figure 1. It might be desirable to describe the method with more precision, e.g., specifically state the cost functions that are minimized during training (particularly for the encoder/decoder; in fairness, the least squares cost is stated), along with any regularization penalties. Significance ------------ I think the paper tackles a highly relevant/significant problem (i.e., learning basis functions for DMD in spectral analysis of the Koopman operator), however, the differences compared to previous work are unclear at best. The modifications to existing algorithms (that of Takeishi) appear minor, and the numerical studies do not evaluate against the method of Takeishi, making it very difficult to say whether the modification is effective or not. Further comments ---------------- I think numerical comparison with Takeishi is quite important/necessary.

Reviewer 2



This work tackles the problem of learning non-linear dynamical systems defined over sets of discrete observations. The idea builds on the theory of Koopman opeators, where non-linear systems can by modelled via linear operators after defining a mapping of the states in an opportune latent space. The authors propose a black-box approximation of this mapping via an auto-encoder-like model. After encoding, a linear operator is learnt on the dynamics of the latent representation. The learnt prediction, as well as the latent representation are subsequently jointly decoded to match the observations. The study provided thorough experimental validation on the problem of fluid-flow modeling on cylinder systems. I am not confident with the problem and thus not aware of the experimental details and complexities. Nevertheless the proposed approach seems to provide competitive results when compared to analogous learning-based methods, while providing accurate predictions and plausible dynamics description. The proposed approach is quite appealing and grounded on nice theoretical basis. Moreover, it is exhibits nice scalability properties. An issue of this work may be related to the interpretability of the modelled dynamics. Since the dynamics are modelled in the latent space, it is rather complicated to grasp any insights on the relationship between the states by inspection of the model parameters. Therefore, interpretability has necessarily to rely on post-hoc simulation and tests. While this may not be an issue in purely predictive tasks, this is clearly a critical aspect whenever we may want to identify meaningful relationship across states. There is also the risk that the dynamics may sensitively differ according to the dimensionality of the latent space. This is to my opinion a key parameter of te model that should be properly tuned and justified.

Reviewer 3



This paper presents an approach to learn the dynamics of the unsteady flow around a cylinder immersed in a fluid in order to use model-based control to reduce the unsteadiness of the flow. The paper has three main parts: 1. The dynamics of the fluid flow are modelled as a linear state-space model operating on a modified state that is the result from applying a non-linear transformation to the actual state of the flow. 2. Model Predictive Control (MPC) is used to reduce flow unsteadiness by actuating the cylinder immersed in the flow. Quadratic MPC can be used because the dynamics are linear in the modified state representation. 3. Insight from the MPC solution is used to design a very simple feedback control law that reduces flow unsteadiness without the need of running MPC. The paper is solid, well written and mostly clear. I would class this paper as a good "application paper" that makes good use of a number of techniques that have been developed in their respective fields. I think that this is a very good paper to introduce Machine Learning techniques to the aerospace engineering and computational fluid dynamics communities. However, I am not sure that it offers useful insights or techniques to the NIPS community. The control part of the paper makes straightforward use of known techniques. The modelling of the dynamics is, in my opinion, more interesting. The basic idea is to learn an auto-encoder that nonlinearly transforms the state in such a way that the dynamics are linear when represented in the encoded state. Section 2.2 is key as it presents the dynamical model and how it is learnt. In my opinion, this section would benefit greatly from explicitly stating what is the loss that is minimised during training. I find Figure 1 nice to get an overall view of what is going on but I also find it slightly ambiguous (although some of the ambiguities are cleared up in the text.) For instance, it could seem that Yhat = decoder(Ytilde) whereas I believe that in actual fact Yhat = decoder(Ytilde_pred). In other words, the element xhat_2 in Xhat is not necessarily the same than the element xhat_2 in Yhat. Something that I have found interesting which I do not remember seeing before is that within this non-standard autoencoder, only the first half the time series is used to compute matrix A via least squares. Presumably, since the model is trained end-to-end, there is backpropagation through the least squares operation and the second half of the time series acts like an internal validation set.